# Physiological Responses to Salt Stress at the Seedling Stage in Wild (*Oryza rufipogon* Griff.) and Cultivated (*Oryza sativa* L.) Rice

**DOI:** 10.3390/plants13030369

**Published:** 2024-01-26

**Authors:** Jacopo Trotti, Isabella Trapani, Federica Gulino, Maurizio Aceto, Miles Minio, Caterina Gerotto, Erica Mica, Giampiero Valè, Roberto Barbato, Cristina Pagliano

**Affiliations:** 1Department for Sustainable Development and Ecological Transition, University of Eastern Piedmont, Piazza Sant’Eusebio 5, 13100 Vercelli, Italy; jacopo.trotti@uniupo.it (J.T.); federica.gulino@uniupo.it (F.G.); maurizio.aceto@uniupo.it (M.A.); erica.mica@uniupo.it (E.M.); giampiero.vale@uniupo.it (G.V.); roberto.barbato@uniupo.it (R.B.); 2Department of Science and Technological Innovation, University of Eastern Piedmont, Viale Teresa Michel 5, 15121 Alessandria, Italy; 3Department of Life and Environmental Sciences, Marche Polytechnic University, Via Brecce Bianche, 60131 Ancona, Italy; m.minio@pm.univpm.it (M.M.); c.gerotto@univpm.it (C.G.)

**Keywords:** *Oryza rufipogon*, *Oryza sativa*, seedling stage, salt stress tolerance, Na^+^/K^+^ homeostasis, C:N ratio, JIP-test, thylakoid membranes, photosynthesis

## Abstract

Domesticated rice *Oryza sativa* L. is a major staple food worldwide, and the cereal most sensitive to salinity. It originated from the wild ancestor *Oryza rufipogon* Griff., which was reported to possess superior salinity tolerance. Here, we examined the morpho-physiological responses to salinity stress (80 mM NaCl for 7 days) in seedlings of an *O. rufipogon* accession and two Italian *O. sativa* genotypes, Baldo (mildly tolerant) and Vialone Nano (sensitive). Under salt treatment, *O. rufipogon* showed the highest percentage of plants with no to moderate stress symptoms, displaying an unchanged shoot/root biomass ratio, the highest Na^+^ accumulation in roots, the lowest root and leaf Na^+^/K^+^ ratio, and highest leaf relative water content, leading to a better preservation of the plant architecture, ion homeostasis, and water status. Moreover, *O. rufipogon* preserved the overall leaf carbon to nitrogen balance and photosynthetic apparatus integrity. Conversely, Vialone Nano showed the lowest percentage of plants surviving after treatment, and displayed a higher reduction in the growth of shoots rather than roots, with leaves compromised in water and ionic balance, negatively affecting the photosynthetic performance (lowest performance index by JIP-test) and apparatus integrity. Baldo showed intermediate salt tolerance. Being *O. rufipogon* interfertile with *O. sativa*, it resulted a good candidate for pre-breeding towards salt-tolerant lines.

## 1. Introduction

Rice (*Oryza sativa* L.) is the second most cultivated cereal crop in the world after wheat, and is one of the major food sources that sustains more than half of the world’s population [1]. Unfortunately, rice is also the most salt-sensitive cereal [2]. Soil salinization is a global threat that affects approximately 7% of the earth’s land surface [3]. This phenomenon is mainly due to the accumulation of soluble salts containing sodium (Na^+^), calcium (Ca^2+^), magnesium (Mg^2+^), chloride (Cl^−^), and sulfate (SO_4_^2−^) ions in the root zone, at concentrations leading to an electrical conductivity (EC) above 4 dS m^−1^ (approximately 40 mM NaCl) and an exchangeable sodium percentage below 15% in soil extracts at 25 °C [4]. Salinity is, instead, the condition occurring to soils with an EC sufficient to cause crop yield reduction, for most of which, rice included, is in the range of 2–4 dS m^−1^ [5,6], a value below that of soil salinization. Soil salinity is becoming a major environmental constraint for rice crop production [7], and will worsen in many regions because of global climate change, especially in the coastal regions and river deltas for ingression of saline cones, and as a consequence of improper irrigation practices and the deterioration of irrigation water [8]. On rice, salt stress exerts its adverse effects mainly at the seedling stage, leading to poor crop establishment, and at the reproductive stage, interfering with grain-filling, thus producing severe yield loss [2,9].

Saline soils usually contain high concentrations of NaCl, which is the salt that predominantly affects plants, inducing osmotic (fast response) and ionic (slow response) stress. NaCl is osmotically active, and reduces the water potential of the rhizosphere producing early water balance problems in the plants, causing water shortages and reduced turgor that affects growth of cells and plant organs. In cereal crops, this effect is major in the leaves, which halt their expansion in a few minutes, then plants partially recover to grow at a reduced rate [10]. Over a longer time, Na^+^ ions, chemically similar to K^+^, can be transported into the plant by systems for K^+^ uptake, and accumulate in the organs, where they exert their toxicity. Na^+^ interferes with the intracellular balance of ions pool and charges, partly at the expense of the K^+^ pool. Since K^+^ is necessary for the activity of many enzymes, and Na^+^ cannot substitute it, high cellular concentrations of Na^+^ or high Na^+^/K^+^ ratios can severely affect various enzymatic processes. Moreover, when K^+^ is displaced by Na^+^, membrane depolarization occurs, leading to K^+^ and water efflux. Therefore, the ability to maintain a lower Na^+^/K^+^ ratio in the cytosol is a successful strategy for salt tolerance, which is mediated by ion transporters and channels and osmolyte production (e.g., proline) [11,12]. This is particularly important in leaves, which are the site of photosynthesis, where high concentrations of salt can induce a rapid decrease in stomatal conductance, followed by a decrease in Photosystem II (PSII) efficiency and an increase in energy loss by heat and in the production of reactive oxygen species (ROS) [13]. In the long-term, exposure to salinity inhibits gaseous exchange and, by damaging the photosynthetic machinery, impairs photosynthesis and inhibits CO_2_ assimilation, mining the stability of carbon and nitrogen metabolism [14,15]. Carbon starvation and oxidative damage induced by excessive ROS formation upon salt stress can concur to premature whole-plant senescence [16], which results in a significant biomass reduction, yield loss, and food quality pauperization, especially in annual crops [17].

Salt tolerance is a complex trait, highly influenced by the environment and large genetic variability for salt tolerance exists among rice (*O. sativa* L.) subspecies and varieties. Based on genetic, agronomic, morphological, and physiological features, *O. sativa* is divided into two subspecies, *indica* and *japonica*, with the first mainly grown in tropical regions, and the second mostly in temperate regions. Both subspecies differ at a varietal level for tolerance to salt stress [18], with *indica* being more tolerant than *japonica* [19,20]. Pokkali, Nona Broka, and FL478 are some of the most salt-tolerant rice cultivars [21] that belong to the *indica* subspecies. The wild ancestor of *O. sativa* is *O. rufipogon* Griff. (also known as Dongxiang wild rice), which has both perennial and annual types [19] widely distributed in the tropics and subtropics of Asia [22]. *O. rufipogon* accessions showed tolerance to some biotic and abiotic stresses [23]. Since this wild rice is a diploid species with the same AA genome as the cultivated *O. sativa*, it has been used for gene transfer to cultivated rice *indica* cultivars in the breeding of disease- and insect-resistant rice [22], and drought- and salinity-tolerant rice [24,25,26,27]. In addition to using wild germplasm, many efforts have been made to develop high-yielding rice cultivars with improved tolerance to salt stress by crossing salt-tolerant with salt-sensitive *O. sativa* cultivars, either *indica* or *japonica*, with conventional and modern techniques [21,28,29]. In the European Union, temperate rice *japonica* cultivars are cultivated in about 410,000 ha, with Italy as the first producer [30]. Previous studies reported a differential salt tolerance among different *japonica* genotypes, and among these are the highly sensitive Vialone Nano and the mildly tolerant Baldo [31].

Considering the great potential of *O. rufipogon* as a possible source of new salinity tolerance genes/alleles useful in future breeding programs, a better understanding of its physiological mechanisms of salinity tolerance, which is still limited [32,33], would be beneficial. Toward this goal, here, we examined the physiological responses to salinity stress (80 mM NaCl for 7 days) in seedlings of the wild *O. rufipogon* accession PI 347745 and the cultivated Italian *japonica* varieties Baldo and Vialone Nano. We report the occurrence of a different tolerance during the early growing stages under moderate salt stress conditions among the three genotypes, with *O. rufipogon* showing a higher survival rate and fewer injury symptoms. The possibilities that the differential tolerance may arise from different water status, Na^+^/K^+^ ion homeostasis, C:N balances, and photosynthetic apparatus performances and integrity were investigated.

## 2. Results

### 2.1. Plant Survival and Growth Capacity 

In order to analyze the response to salt stress by rice at the seedling stage, rice seedlings were grown in hydroponic culture for 2 weeks; during the second week, seedlings were exposed to 80 mM NaCl, which is considered a suitable concentration for studying the salt tolerance of rice grown in hydroponics over this time interval [34].

After 7 days of salt exposure, *Oryza rufipogon* (hereafter, OR), *Oryza sativa* var. Baldo (B), and *Oryza sativa* var. Vialone Nano (VN) plants were evaluated by visual assessment for salt stress injuries, based on the SES score (according to IRRI Standard Evaluation System [35]) routinely applied to test rice salt tolerance [36] (Figure 1A and Appendix A). This analysis showed a high variation in the percentage of plants with SES score values of 7–9, corresponding to dying or dead plants, respectively. This percentage was lower in OR (23%) than B (34%) and VN (45%). The percentage of plants that showed no injury symptoms (SES score of 1) was much lower, attesting to 15% in OR, 5% in B, and 9% in VN. After salt treatment, the majority of plants (62% of OR, 61% of B, and 46% VN) showed few stress symptoms, such as whitish and rolled leaves and reduced growth and tillering, variable in extent (corresponding to SES score values of 3–5). The evaluation of rice for salinity tolerance based on phenotypic traits indicated OR as the most tolerant, with 77% plant survival with no to moderate injury symptoms (SES score values of 1–5), followed by B (66%) and VN (55%).

The most representative plants for salt treatment that were still alive, i.e., those with SES 3–5, were used for all subsequent analyses comparing them to control plants. Only for biochemical analyses, treated plants with SES 1 were also included in the sampling to reach an amount of fresh leaf material sufficient to perform the thylakoid extraction and subsequent analyses.

To evaluate how salt treatment affected plant growth, we analyzed different plant growth parameters relative to the aerial part (shoot length, number of leaves, length, and area of each leaf) and root system (total length and volume, average diameter, surface area, tips number, and branching degree) of OR, B, and VN seedlings.

In control conditions, OR and VN were bigger in size than B, as attested by the longer shoot length (Figure 1B) and root system length (Table 1). After salt treatment, all the three genotypes significantly reduced their shoot length to about 1/3 of control in OR and 1/4 in B and VN (Figure 1B), but showed different behaviors for the root system. Despite a common slight increase in the maximal root length (Figure 1B), OR and VN significantly reduced the length of the total root system to about 1/2 and 1/3 of control, respectively, while B kept it almost constant (Table 1 and Appendix A). Under salt stress, the total root system length was similar in the three genotypes, ranging between 140 and 160 cm. Similarly, the closely related root size parameters volume and surface area also showed no significant differences among genotypes, with volume ranging between 0.14 and 0.18 cm^3^ and surface area between 17 and 21 cm^2^ (Table 1). On the contrary, salt induced a variation in the average root diameter significant only in OR (+14% respect to control), despite similar dimensions in the range of 0.348–0.372 mm that were observed in the control condition in the three genotypes. Without salt, OR and VN had a similar amount of root tips, which was higher (+30%) than in B; after salt treatment, this parameter was reduced of at least 1/3 in OR and VN, but was kept almost constant in B. Because of the same trend of variation in root length and number of root tips in response to salt, the derived branching degree parameter also did not change significantly among the genotypes (Table 1).

Without salt, the three genotypes after two weeks formed four leaves, which were much bigger in both surface area (Figure 1C) and length (Appendix A) in OR, followed by VN, while B had the smallest leaves. After salt treatment, only B displayed the emergence of a fourth small leaf, whereas all the genotypes reduced the size of all their leaves, although to different extents. OR and VN reduced the size of their leaves more than B when compared to their controls (OR −54%, VN −42%, B −18% for total leaf area; OR −38%, VN −28%, B −19% for total leaf length). Despite the higher percentage of relative reduction in leaf size in OR, this genotype under salt showed a superior photosynthesizing area of the most fully expanded and youngest leaf (the third) with respect to the other genotypes (OR 3.74, B 2.59 and VN 3.10 cm^2^).

The measure of the relative water content (RWC) on the third leaf displayed a severe water reduction in VN (−26%), and the onset of a water deficit in B (−12%), while in OR, no significant water loss was recorded (Figure 1D).

Plant exposure to the saline solution resulted in significant reductions in shoot and root fresh and dry weight (Figure 2). Relative to the control, the magnitude of decrease in fresh biomass was similar both for roots and leaves in OR (−61% and −60%), but was about twice in leaves than in roots in B (−41% and −20%) and VN (−46% and −25%) (Figure 2A). This led to a reduction in the shoot/root fresh weight ratio in salt-treated plants with respect to their control counterparts, which was higher in VN than B, whereas in OR, this ratio remained constant (Figure 2B). On a fresh biomass basis, under salt stress, OR reduced its size, maintaining the biomass partitioning between roots and shoots of the control, while VN and B changed biomass partitioning, favoring the expansion of the root system more than the shoot system. Considering dry biomass (Figure 2C) with respect to fresh biomass (Figure 2A), OR and B Ctrl plants decreased similarly in the matter of both roots and leaves (weight reduction of about 80–90%), denoting an initial similar water content for both organs. The lower biomass reduction (weight reduction of about 65–75%) observed in VN Ctrl plants, especially in roots, suggests a lower hydration level for this genotype. After salt treatment, OR reduced the dry weight of roots and leaves in similar proportions (−34% and −33%, respectively), B reduced only leaf dry weight (−31%), while VN reduced both root and leaf dry weights to different extents (−31% and −24%, respectively) (Figure 2C). After salinization, the observed reduction in total leaf dry weight in the order VN < B < OR (Figure 2C) was in accordance with the (third) leaf relative water content value following the reverse order OR > B > VN (Figure 1D).

### 2.2. Na^+^ and K^+^ Homeostasis in Roots and Leaves 

Regardless of the genotype, salt treatment induced a general increase in sodium (Na^+^) (Figure 3A) and a decrease in potassium (K^+^) (Figure 3B) in both roots and leaves. In salt-treated plants, B and VN contained, respectively, 3-fold and 4-fold more Na^+^ in leaves than in roots, while OR had a similar content of Na^+^ in the two organs (Figure 3A). Moreover, OR translocated less Na^+^ to the photosynthetic tissue with respect to the two cultivated genotypes (56% and 61% less than B and VN, respectively). Conversely, in salt-treated plants, the K^+^ content decreased in a common way in the three genotypes, by 60% in the roots and by 10–20% in the leaves (Figure 3B). Consequently, after salt stress, the Na^+^/K^+^ ratio in the roots was about 1.6-fold higher in B and VN with respect to OR, and in the leaves, it was significantly increased in the order VN > B > OR (Figure 3C). From the measurements of sodium and potassium content, when compared to the cultivated rice varieties, OR resulted as more able to accumulate Na^+^ at the root level, reducing the translocation to the leaves, where it showed the lowest Na^+^/K^+^ ratio.

### 2.3. Carbon and Nitrogen Content in Leaves 

The C and N quota in leaves of OR, B, and VN plants grown in control conditions or that were NaCl-treated was quantified as the percentage on dry weight (Figure 4A). The analysis showed that OR samples had, on average, a percentage of C of about 40.5%, while N was roughly 3.5%, with no major effect from the salt treatment. In B control plants, the C percentage was about 41.8%, slightly higher compared to OR, while in NaCl-treated B plants, the C percentage decreased significantly to about 38.8%. N percentage in control B plants was similar to that of OR, and in the NaCl-treated B plants, it was slightly decreased. From these analyses, VN was the genotype most affected by salinity, as in VN, the C percentage significantly dropped from about 38.0% in the controls to about 33.5% in the NaCl-treated plants and, similarly, the N percentage decreased from about 3.5% to roughly 3.0%. Despite the differences in the quota of C and N detected in some of the samples, the C:N ratio remained almost unaffected (Figure 4B). Similarly, the C stable isotope composition (δ ^13^C), a parameter that became more negative within samples more depleted in the heavier isotope ^13^C, did not show major changes as a consequence of the salinity (Figure 4C), although OR was the genotype showing the most negative average values, and in VN, the NaCl-treated plants had an average value slightly less negative compared to the respective control.

### 2.4. Pigments Content and Photosythetic Performance by JIP-Test

Regardless of the salt treatment, the youngest, most fully expanded and metabolically active leaf was the third in all genotypes (Figure 1C). On this leaf, we performed analyses of the pigment content (Table 2) and OJIP chlorophyll *a* fluorescence, with calculation of derived main JIP-parameters (Figure 5 and Figure 6, Appendix A).

In control conditions, OR showed significantly higher amounts of chlorophyll *a* (Chl *a*), chlorophyll *b* (Chl *b*) and carotenoids than B and VN (Table 2). After salt treatment, OR reduced all these pigments of about 1/3, and B showed a tendency to slightly increase the amount of Chl *a* and Chl *b*, while VN kept constant the total chlorophylls and slightly decreased carotenoids. In general, after salt treatment, the Chl *a*/*b* ratio did not significantly vary in the three genotypes, while the Chl/Carotenoids ratio significantly increased in B and VN (Table 2).

In Figure 5A, the typical chlorophyll *a* fluorescence transients recorded on dark-adapted leaves are shown for control and salt-treated plants. In all cases, some modifications in the curves were observed. In order to emphasize differences between the relative maximum variable fluorescence upon salt treatment, a first round of normalization at F_o_ was performed (Figure 5B). Clearly, the most pronounced effect of salt was observed in VN, whereas in B, the effect was barely detectable. An intermediate effect was instead observed in OR. Finally, a double normalization was performed at F_o_ and F_m_, allowing for plotting transients as relative variable fluorescence (V_OP_) (Figure 5C). In this case, the distortion of the shape of the transients induced by salt treatment was evident, as highlighted in the inset of Figure 5C, where the parameter ΔV_OP_ was calculated as the difference between the relative variable fluorescence curves of NaCl-treated and control plants. Differences among transients may be further highlighted by normalization at different time points, as well as plotting the corresponding difference kinetics. In Figure 5E, data were normalized between O (20 μs) and J (2 ms), and presented as V_OJ_ = (F_t_ − F_o_)/(F_J_ − F_o_). The corresponding ΔV_OJ_ (inset of Figure 5E), resulting from subtraction of the control from the respective salt-treated sample, highlighted the presence in all genotypes of the K-band, with slightly higher amplitude in OR than B and VN. Normalization of data between O (20 μs) and K (300 μs), followed by calculation of corresponding ΔV_OK_, allowed the resolution of the L-band (Figure 5D). Again, the effect of salt treatment was slightly more pronounced in OR than in VN and B (see inset of Figure 5D). Normalization of data between I (30 ms) and P (300 ms), followed by calculation of corresponding ΔV_IP_, showed a common positive band with a peak between 80 and 110 ms, with amplitude 3-fold higher in VN with respect to B and OR (Figure 5F).

All the parameters derived from the Chlorophyll *a* fluorescence curve are presented in Figure 6 as the relative variation in salt-treated samples with respect to the corresponding control samples (instead, Appendix A contains the original data for each JIP-parameter shown in Figure 6). After salt treatment, VN showed a higher reduction in the F_o_ and F_m_ values with respect to the other genotypes, and OR and B had a similar reduction in F_o_, whereas OR had a reduction in F_m_ higher than B. The area above the fluorescence curve between F_o_ and F_m_ is proportional to the pool size of the electron acceptors Q_A_ on the reducing side of PSII. When the electron transfer from the PSII reaction center to the quinone pool is slowed down, this area is reduced. The Area parameter in salinized plants relative to the control decreased by 45% in VN and 27% in OR, while it was unaffected in B. In salt condition, only OR showed a significant increment of the M_o_ value (expressing the fractional rate of closed reaction centers accumulation) and decrement of the proportion of active PSII RCs (calculated as [RC/(RC)_Ctrl_] = (RC/ABS)_NaCl_/(RC/ABS)_Ctrl_ according to Tsimilli-Michael, 2020 [37]). Upon salt stress, the F_v_/F_o_ parameter, which accounts for the simultaneous variations in F_m_ and F_o_ in determining the maximum quantum yield of PSII (ϕP_o_), and estimates the efficiency of the water-splitting complex on the donor side of PSII, was greatly lowered in VN (−36%), while the ϕP_o_ decreased slightly only in VN (−6%) relative to control.

Regarding the specific energy fluxes expressed per reaction center (RC), upon salt stress, a generally similar small increase was noticed in all genotypes for the photon absorption (ABS/RC), the electron trapping efficiency (TR_o_/RC), the electron transport within the reaction center (ET_o_/RC), and the flow of electrons further than PSII (RE_o_/RC). Concomitantly, an increase in the activity of energy dissipation (DI_o_/RC) was observed markedly in VN (+47%) and, to a lesser extent, in OR (+30%). The Ψ_o_ and φE_o_ decreased slightly only with salt treatment in OR plants as compared to control.

The photosynthetic performance index on an absorption basis (ABS) PI_(ABS)_, an indicator of sample vitality, was one of the most affected parameters observed in the JIP-analysis of salinized rice plants. The PI_(ABS)_ is a multifactorial parameter that takes into account the fraction of active PSII reaction centers per chlorophyll, the maximal energy flux that reaches the PSII reaction centers, and the electron transport at the onset of illumination. A drop of 40% in the PI_(ABS)_ was observed in VN plants in the saline condition as compared to control plants, whereas in OR, this reduction was 25%, and 8% in B.

### 2.5. Thylakoid Membranes and Photosystem I and Photosystem II Protein Quantification

The electrophoretic pattern of thylakoid membranes isolated from salt-treated and control plants revealed a generally similar composition among the three genotypes, with some evident differences in the abundance of proteins in the region below 20 kDa in OR compared to the other two genotypes, independent of the salt treatment (Figure 7A).

The salt effect on the expression of main proteins of the Photosystem I (PSI) and Photosystem II (PSII) complexes of the thylakoid membranes was semi-quantitatively determined with Western blotting (Figure 7B) followed by densitometry analyses (Figure 7C). The results indicated that after salt treatment, the amount of the PSI reaction center PsaA subunit increased significantly in VN plants (2.5-fold as compared to control), and only slightly in B plants (1.4-fold as compared to control), whereas it remained almost constant in OR plants (NaCl/Ctrl PsaA ratio close to 1). Salt treatment had no major effect on the expression of the PSII reaction center proteins D1 and D2 and the inner antenna proteins CP47 and CP43 in OR plants (NaCl/Ctrl protein ratios close to 1), whereas a general modest increase was observed for these proteins in B and VN plants (NaCl/Ctrl protein ratios ranging between 1.2–1.5). The salt treatment did not significantly affect the abundance of the major LHCII proteins in all the genotypes. Conversely, the effect of salt on the expression of the Oxygen Evolving Complex (OEC) subunits PsbO, PsbP, PsbQ, and PsbR was genotype-dependent. First, it should be noted that the Western blotting with antibody against PsbQ protein revealed a lower intensity signal, regardless of the salt treatment, in OR plants compared to the other two genotypes (Figure 7B), which was not observed with antibodies for the other OEC subunits. The lower abundance of the PsbQ (17 kDa) protein detected in the wild rice OR is in agreement with the lower amount of proteins in the molecular mass region below 20 kDa observed in the corresponding SDS-PAGE of thylakoid membranes shown in Figure 7A. Salt treatment induced a general increase in the PsbP and PsbQ proteins in OR plants (respectively, 1.7- and 1.3-fold relative to the control); of PsbO, PsbP, and PsbR in VN plants (NaCl/Ctrl protein ratios ranging between 1.4–1.5); and of PsbP and PsbR in B plants (respectively, 1.2- and 2.2-fold relative to the control). Only a small reduction in the PsbQ protein (0.8-fold relative to the control) was detected in salt-treated B plants.

## 3. Discussion

At present, salinity is the most predominant hindrance to rice production after drought, an outcome related to the fact that rice is the most sensitive cereal to salt stress, with an average EC threshold of 3.0 dS m^−1^ [39]. Here, we performed hydroponic experiments imposing a moderate salt stress (80 mM NaCl, corresponding approximately to an EC of 8 dS m^−1^) for 7 days during the early seedling stage (2–3 leaves), which is one of the developmental stages at which rice plants are more sensitive to salt [36]. Two Italian *O. sativa japonica* cultivars with contrasting behavior towards salt, the mildly tolerant Baldo and sensitive Vialone Nano [31], were compared to the wild *O. rufipogon*, which seems to possess superior salt tolerance [33]. Since an in-depth physiological characterization for this wild rice is still missing, we undertook it to understand if *O. rufipogon* could effectively serve as source of new genes for pre-breeding towards salt-tolerant lines.

### 3.1. Effects of Salinity on Plant Survival

Symptoms of stress injuries at the early seedling stage manifested in all genotypes, first on the oldest (first and second) leaves, and finally on the youngest (third) leaf, leading to progressive rolling, whitening, and subsequent death. From a visual evaluation of symptoms of salt stress injuries, based on the SES score [35] used for mass screening (Figure 1A and Appendix A), OR showed higher survival rate and was the most tolerant genotype. On the contrary, between the two Italian cultivars, VN was the most sensitive, in accordance with previous results obtained in similar salt treatments under hydroponics [31,34].

### 3.2. Effects of Salinity on Plant Growth, Water Status and Na^+^ and K^+^ Ion Homeostasis

Under control conditions, the three genotypes showed different root and shoot sizes; specifically, OR and VN displayed bigger sizes than B, with OR > VN for the root length parameter (Figure 1B and Figure 2C, Table 1). Salinity induced a reduction in plant growth in all the genotypes, even though to different extents (Figure 1, Figure 2 and Appendix A, Table 1) and with different levels of impairment to the growth of shoots and roots in the three genotypes. Indeed, despite a common similar reduction in the maximum shoot length (Figure 1B), the extension of root system (i.e., total length, volume and surface area) was kept constant in B, while it was reduced in the genotypes with longer roots (Table 1). A similar reduction in root network length was reported in different *O. sativa* cultivars under salt stress at the seedling stage, especially in bigger ones [40], and it was linked to suppression of cell division and elongation [41]. Development of lateral roots producing higher root branching can influence root hydraulic conductivity and water supply [42]; thus, it can be an important trait for plants growing under salt stress. Divergent results exist regarding the effects of salt stress on the development of the lateral roots [43]. In our experiments, however, we did not notice an effect of salinity on the branching degree, instead we noticed an effect on root elongation, which was reduced especially in the rice genotypes with longer roots (Table 1), and was paralleled by higher reductions in fresh (Figure 2A) and dry (Figure 2C) root weights.

Under salt stress, shoot growth is generally more affected than root growth [44], and can lead to changes in the allocation of biomass between roots and shoots, which is a recognizable indication of salinity stress [18,45]. We found that, upon salt stress, the relative decrease in fresh biomass of the whole plant was in the order OR > VN > B (Figure 2A); however, it was differentiated between leaves and roots among the three genotypes. OR plants decreased the biomass of both roots and leaves to a similar extent, while B and VN reduced the biomass of the aerial part more, rather than the root system. Consequently, upon salt stress, the shoot/root fresh weight ratio remained almost unchanged in OR, whereas it decreased significantly in B and, to a higher extent, in VN (Figure 2B). Under analogous salt treatment, a similar high reduction in shoot/root fresh biomass ratio was observed in VN, and in the salt sensitive *japonica* Onice rice cultivar [34], while the tolerant *indica* IR64-*Saltol* cultivar kept this ratio almost constant. Shoot growth impairment and changes in biomass partitioning induced by salinity have been shown to be linked to variations in hormonal concentrations of auxins and cytokinins in roots and shoots [45]. The unchanged shoot/root ratio in OR allows this wild rice to keep the original mass partitioning and overall plant architecture under salinity, while reducing its size, which might be determinant for obtaining an advantage in terms of higher survival rate.

A decrease in the shoot/root ratio has been related to factors associated with water stress (osmotic effect), rather than to a salt-specific ionic effect [46]. Under salt stress, an important challenge to plant functioning is the ability to maintain water content in tissues at optimal levels. The loss of water from tissues can lead to rapid changes in cell expansion and division, the accumulation of abscisic acid, and stomatal closure [46]. Most of these effects can become evident with very small reductions (<10%) in tissue water content. In line with the observed shoot/root variations (Figure 2B), upon salt treatment, the hydration status of rice plants, assessed by measurement of leaf RWC, was optimal in OR, slightly compromised in B, and severely affected in VN (Figure 1D). Also, Formentin et al. [47], by using a similar salt treatment, found a severe leaf RWC reduction in VN seedlings, but not in B rice plants, probably because of the shorter duration of salt treatment. Since, under salinity, a higher RWC correlates with a greater control of the stomatal aperture [47], our results might suggest a superior control of stomatal opening in OR plants, which can contribute to a more balanced transpiration rate able to reduce the salt load to leaves (Figure 3A), while improving plant growth and survival (Figure 1A). Accordingly, OR plants also retained a higher leaf area, especially of the youngest third leaf, with respect to the other genotypes (Figure 1C and Appendix A), despite the general leaf growth reduction induced by salt, which affects the cell expansion especially of young leaves in rice [48].

The ability of plants to maintain shoot Na^+^ content below non-toxic levels is vital to achieve better plant growth and a higher survival rate under saline conditions. Root apoplastic bypass is the major route of Na^+^ entry in rice [49,50]. Plants adopt different strategies to cope with excessive Na^+^, either by sequestering Na^+^ into vacuoles or by retrieval of Na^+^ from the transpiration stream through specific transporters to prevent Na^+^ transfer to young leaf blades [12,51,52]. These methods allow plants to keep the cytoplasm free from Na^+^ toxic concentrations, particularly in the mesophyll cells, maintaining ion homeostasis in terms of a relatively low Na^+^/K^+^ ratio that is considered a critical factor for determining plant salinity tolerance [53,54]. In salinized conditions, B and VN plants accumulated more Na^+^ in leaves than roots, while OR retained a higher amount of Na^+^ in the roots with respect to the other genotypes, thus limiting its translocation to leaves (Figure 3A), which ultimately contributes to lower the Na^+^/K^+^ ratio therein (Figure 3C). Roots are more tolerant to salt than leaves [12], and storing salt in the roots for as long as possible can be a successful strategy, as also observed in other wild rice species [55]. It is worth noting that the wild OR was the only genotype that significantly increased the root diameter under salinity (Table 1). This feature might be an anatomical adaptation at the root level in this wild OR accession, possibly related to succulence of the cortex [56] which, by increasing the volume of vacuoles, allows the accumulation of higher quantities of ions and water in the roots [10]. An analogous strategy has been found at the leaf level in a OR line after a prolonged salt treatment (up to 6 weeks) by Solis et al. [33], conferring a higher capacity of vacuolar Na^+^ sequestration in this wild rice. Thus, differently from the cultivated genotypes, in the wild OR rice, Na^+^ might act as a “cheap” inorganic osmoticum either at root or at leaf level, depending on the duration of the stress. In conclusion, differences in sodium allocation and better Na^+^ and K^+^ homeostasis in the leaves might explain the higher tolerance of OR to ionic stress.

Carbon and nitrogen are the two most abundant nutrient elements in plants, and their metabolisms constitute two metabolic pathways fundamental for plant growth that are highly interconnected, such that the ability of plants to take up and utilize N may be compromised if C availability is insufficient [57]. Salt stress can inhibit either carbon uptake as CO_2_ in the leaves or nitrogen uptake, mainly as nitrate in the roots, and further metabolisms in the whole plant, especially in the older leaves in rice [58]. Salt stress had no impact on the %C and %N content in OR leaves, while it triggered a significant reduction in both elements in B and VN (Figure 4A), paralleled by the increment of sodium (Figure 3A), attesting to an alteration in carbon and nitrogen uptake and assimilation in the two cultivated genotypes, more pronounced in the salt sensitive VN. In addition to the cell quota of C and N, plants are known to discriminate against the heavier C stable isotope, ^13^C, during the photosynthetic uptake of CO_2_. The C stable isotopic composition has, likewise, been suggested as a useful trait to be characterized when evaluating the salt tolerance [59,60]. In C3 plants such as rice, the δ ^13^C values are usually in the range of −21‰ to −35‰ [61]. The depletion of ^13^C in the bulk leaf material may vary as result of multiple processes, primarily the activity of carboxylases. Rubisco strongly discriminates against ^13^C, leading to a fractionation of carbon stable isotopes of about 29‰ [62]. In addition to the effects of carboxylases and (although to a lower extent) of several other metabolic enzymes on the ^13^C abundance in plants, the diffusion of CO_2_ through stomata into leaves may also account for a 4.4‰ C stable isotopes fractionation [62]. Therefore, any factor affecting photosynthesis or stomatal conductance may affect the δ ^13^C of the leaf sample [61]. Consistently, although we detected only small differences in the δ ^13^C of the controls of the three genotypes, among the salt-treated plants, the most sensitive genotype (VN) showed a tendency to a less negative δ ^13^C, i.e., a lower discrimination against ^13^C (Figure 4C). This suggests VN will be affected by a higher impairment of stomatal conductance and/or of photosynthesis. The increase in δ ^13^C in salt-treated VN plants is also in line with the higher δ ^13^C values observed in drought-stressed C3 plants of the Mediterranean region during the dry season and, more generally, in water-stressed plants [63].

### 3.3. Effects of Salinity on Plant Photosynthetic Apparatus Integrity and Performance

As other wild rice species [33,55], OR possesses a higher chlorophyll content than cultivated rice genotypes (Table 2). After salt treatment, in the more metabolically active leaf (the third), OR showed a major reduction in photosynthetic pigments, leading to a content similar to those of B and VN plants which, conversely, did not significantly change their amounts. However, it should be noted that the pigments’ content has been normalized on a leaf fresh weight basis, which was subjected to a different decrement of the RWC in B and VN after salt treatment (Figure 1D). Indeed, even though B showed a tendency to increase the pigment content (Table 2), it was counteracted by the reduced water content of the leaf (Figure 1D). Based on the same consideration, in VN, an effective reduction in pigments occurred. Generally, salt-tolerant species show increased or unchanged chlorophyll content under salinity, whereas salt-sensitive species decrease chlorophyll levels. In rice, however, there is great variability among salt-sensitive and tolerant cultivars in terms of photosynthetic pigments’ retention, and not always tolerant varieties show negligible reduction in pigment contents [54,64], since large genotypic differences in the total chlorophyll content exist [64]. Indeed, in some cases, the retained proportion of green leaves relative to total leaves might correlate better with the survival rate [64] than the chlorophyll content measured in a specific leaf/portion of leaf does, as also happens in this study (Figure 1A).

Photosynthesis is the most important physiological process in plants, which can be affected by different environmental stresses. The measurement of photochemical processes by chlorophyll *a* fluorescence induction curves allows the evaluation of the physiological condition of PSII and the photosynthetic electron transport chain components, for obtaining an idea of the intensity of the stress encountered by the plant [65]. Among the three genotypes, under salt stress, VN showed the highest impairment of the PSII functionality, as evident from the major changes in the shape of OJIP transients (Figure 5), while B was the least affected. In VN, salt induced a strong decrease in F_o_, F_m_, and F_v_ (Figure 5A,B and Figure 6, Appendix A). The strong decrease observed in F_m_ at the P-step may be due to a curtailment in the electron donation from the PSII donor side, which results in the accumulation of P680^+^ [66], and to a decrease in the pool size of Q_A_^−^. The strong reductions in F_v_/F_o_, which indicate a loss of efficiency of the water-splitting reaction on the donor side of PSII, and of the Area above the fluorescence induction curve, which is proportional to the pool size of the reduced Q_A_ on the reducing side of PSII, suggest that salt stress inhibits the electron transfer rate at the donor side of PSII in VN rice, as in wheat [67].

More insight on the impact of salt stress on PSII can be obtained from some key parameters of the JIP-test (Figure 6 and Appendix A). Notably, our results showed an increase in the ABS/RC ratios under salt stress in all genotypes, which might be due to an inactivation of some PSII RCs, rather than to an increase in the antenna size. Indeed, significant changes were detected neither in the leaf Chl *a*/*b* ratio (Table 2), nor in the abundance of the LHCII in the thylakoids (Figure 7). The decrease in the active RCs under salt stress might prevent a decrease in the electron transport flow (ET_0_/RC), ultimately favoring the excess energy dissipation (DI_0_/RC) in order to limit the photo-damage to the photosynthetic apparatus. Two important JIP-test parameters used for estimation of plant vitality under several abiotic stresses are the F_v_/F_m_ (ϕP_o_), reflecting the PSII primary photochemistry, and the performance index (PI) [65]. PI_(ABS)_ is an integrative parameter that takes into account the amount of reaction centers per chlorophyll (RC/ABS), a parameter related to primary photochemistry (ϕP_o_) and a parameter related to electron transport (Ψ_o_) [68]. In salt-stressed rice, the F_v_/F_m_ parameter was slightly reduced only in VN, while it was unaffected in the other genotypes; conversely, the PI_(ABS)_ decreased sensibly in all the genotypes, although to a different extent (Figure 6). Based on these results, the PI parameter was more sensitive than F_v_/F_m_ for evaluating the state of the photosynthetic apparatus and performance of PSII under salt stress, as also shown in other studies [13,64,67,69,70]. Based on this parameter, VN showed the highest impairment of the PSII functionality, which was less affected in B and OR.

Analysis of chlorophyll fluorescence transients showed some modifications in major steps of the OJIP curve under salt treatment (Figure 5C), whose interpretation may help to understand the mechanism of damage to the photosynthetic process. In particular, NaCl affected the rate of primary photochemistry observed as an increase in fluorescence at the O-J phase slightly higher in OR, with the appearance of a small K-band (Figure 5E), indicating a perturbation during the water decomposition at the OEC [65]. In addition, the presence of an L-band in all salt-treated samples (Figure 5D) indicated a loss of connectivity among PSII units in all genotypes upon salt stress [37]. A compensative response in OR might be by increasing the amount of the OEC PsbP and PsbQ subunits, which are involved in the modulation of the availability and optimization of the concentrations of Ca^2+^ and Cl^−^ ions essential for PSII oxygen evolution activity [71,72], whose uptake is impaired under salt stress, since Ca^2+^ is reduced and Cl^−^ is increased [9]. It is noteworthy that PsbP and PsbQ (and, slightly, PsbR) are the only subunits of the photosynthetic apparatus found to be modulated by salt in OR (Figure 7C) which, in general, retained its overall integrity in this plant. NaCl in VN plants induced an increase in the relative amplitude of the I-P phase of the fluorescence curve (V_IP_) (Figure 5F), which is generally attributed to the reduction in the electron transporters (ferredoxin, intermediary acceptors, and NADP) at the PSI acceptor side [65]. The V_IP_ was suggested to be related to the content of the PSI reaction centers in leaves [73]. Accordingly, VN plants under salt stress displayed a higher V_IP_, paralleled by an increase in PSI (PsaA reaction center subunit) content (Figure 7B,C). The increment of PSI reaction center subunits has been already detected in rice under salt stress [74]. When the PSII activity is compromised (Figure 6), and the net photosynthesis is reduced in salt-treated rice [75], a higher quantity of PSI might suggest an enhanced cyclic electron flow around PSI for the photo-protection of PSII and for balancing the ATP/NADPH production, as already observed under stressful conditions like salt and drought [76,77]. Under salt stress, conversely from OR, B and VN increased the synthesis of the PSII reaction center subunits D1, D2, CP43, and CP47 (Figure 7B,C), as also seen in other glycophytes [78,79]. This might be a strategy during the PSII repair-cycle to maintain PSII functionality, which was successful only in B, where the damage induced by salt was limited. As in OR, B and VN also showed a high modulation of the OEC subunits (Figure 7B,C), suggesting their high sensitivity to salt stress.

Photosynthetic proteins play a major role in regulating photosynthetic electron transport and, ultimately, plant fitness. In response to salt stress, OR slightly increased the abundance of some PSII OEC subunits, and only partially decreased the PI_(ABS)_; B mostly increased the abundance of several PSI and PSII RC subunits, retaining a stable PI_(ABS)_; conversely, VN, despite a major change in the abundance of most of the PSI and PSII subunits, showed the highest decrease in the PI_(ABS)_. Figure 8 summarizes the major effects of salt stress on the modulation of the photosynthetic energy fluxes per RC, performance index, and abundances of PSI and PSII subunits in the three rice genotypes. It is evident that the acclimation of the photosynthetic apparatus was a strategy advantageous for salt-tolerance in OR and B, but not in VN, where it was not sufficient to counterbalance the major detrimental effects of salt on key aspects of plant’s physiology, like growth capacity and water and ionic relations heavily compromised in this genotype.

## 4. Materials and Methods

### 4.1. Plant Material

Rice (*Oryza sativa* L., cv Baldo and Vialone Nano and *Oryza rufipogon* Griff. accession PI 347745) seeds were washed with 70% (*v*/*v*) ethanol, rinsed in distilled water, and surface-sterilized with 1% (*v*/*v*) sodium hypochlorite for 20 min, rinsed three times in water, and germinated on moistened filter paper for 7 days at 24 °C. Germinated seedlings were transferred to pots, and grown hydroponically in Long Ashton nutrient solution [80] for the first 7 days, followed by another 7 days with or without addition of salt, supplied as NaCl at a final concentration of 80 mM. Growth solutions were renewed every three days. Plants were grown in a growth chamber with the following conditions: day/night temperature, 22/26 °C; relative humidity, 60%; photoperiod, 16/8 h; light intensity, 150 µmol photons m^−2^ s^−1^. Plants were grown for two weeks in hydroponic culture, and then harvested for the following analyses.

### 4.2. SES Evaluation and Plant Selection

The stress injury score based on the IRRI standard evaluation system [35] was recorded on each plant 7 days after the salt treatment. At least three independent experiments were performed (three replicates of 30 plants each). After the evaluation, dead or almost dying plants (i.e., with SES scores of 9 and 7, respectively) were excluded from all subsequent analyses.

### 4.3. Biomass and Morphometric Measurements 

At harvest, at least 10 NaCl treated plants with SES scores of 3 and 5 displaying injury symptoms, and corresponding control plants, were divided into roots and shoots, and both were measured in length. Leaves were cut from shoots and further fresh weighed, as well as the root system. The leaves and the root system were digitized with a scanner and analyzed with WinRhizo Pro V 2002c software (Régent Instruments, Québec City, QC, Canada) for determination of the leaf area, leaf length, total root length, root surface area, total root volume, root average diameter, and number of root tips.

After oven-drying for at least 5 days at 50 °C, the leaf and root dry weights of each harvested plant were measured. Dried samples from different plants were pooled together, ground to a fine powder, and then used for determination of ion content and carbon and nitrogen content.

### 4.4. Relative Water Content

The third leaves were cut from the plants and immediately weighed (FW). Then, the leaves were sealed inside falcon tubes, with their petiole submerged below deionized water and kept in the dark for 24 h. After rapid removing of the excess water on the surface using paper towels, the samples were weighed to obtain the turgid weight (TW). Dry weight (DW) was measured after oven-drying for at least 5 days at 50 °C, until the weight remained constant. The relative water content (RWC) of the leaves was determined according to [81] as:RWC (%) = [(FW − DW)/(TW − DW)] × 100.

### 4.5. Determination of Ion Contents in Roots and Leaves 

Aliquots of ca. 0.5 g of dry roots and leaves samples were digested with 2 mL of 69% nitric acid, 1 mL of 30% hydrogen peroxide and 5 mL of high-purity water inside PTFE vessels, using a Milestone (Sorisole, Italy) START D model microwave oven. High-purity water with resistance > 18 MΩ·cm was obtained with a Milli-Q (Milford, MA, USA) apparatus. TraceSelect 30% hydrogen peroxide and 69% nitric acid were purchased from Fluka (Milan, Italy). Elements stock solutions (Inorganic Ventures, Lakewood, NJ, USA) were used for external calibration and internal standardization.

Determination of Na^+^ and K^+^ was carried out with a Thermo Scientific (Waltham, MA, USA) iCAP^TM^ RQ inductively coupled plasma mass spectrometer with single quadrupole technology. The instrument is equipped with an ESI (Omaha, NE, USA) PFA 100 MicroFlow nebulizer, a Peltier-cooled quartz spray chamber operating at 3 °C, a 2.0 mm ID quartz injector, and a demountable quartz torch. Measurements were carried out exploiting an ESI (Omaha, NE, USA) SC-4 DX autosampler. To overcome the spectral interferences, the Collision Cell Technology (CCT) was used with He gas at 3.5 mL/min and a kinetic energy discrimination (KED) barrier of 2 V. The instrument and accessories were PC-controlled by Qtegra^TM^ v. 2.10.4345.136 software. Instrumental parameters were as follows: forward power, 1550 W; plasma gas flow, 14.0 L/min; nebulizer gas flow, 0.9 L/min; auxiliary gas flow, 0.8 L/min. Three replicates were made for a total acquisition time of 30 s. The following isotopes were used: ^23^Na, ^39^K. All samples were diluted 1:100 with ultrapure water before analysis.

Interference due to oxide formation was evaluated as follows: CeO^+^/Ce^+^ < 0.5% in KED mode. A stability test performed before each session by monitoring ^7^Li, ^59^Co, ^115^In, ^140^Ce, and ^238^U yielded a precision higher than 2%. The instrumental precision was better than 2%, while the overall precision, involving both the sample preparation and instrumental analysis, was better than 5%, as calculated on five genuine replicates. Background signals were monitored at 5, 101 and 220 *m*/*z* to perform a sensitivity test on the above-reported analyte masses. CCS-4 multi-element standard solution from Inorganic Ventures (Christiansburg, VA, USA) was used to prepare 1 and 0.1 mg/L solutions in 1% nitric acid. Internal standardization monitoring ^115^In was used to correct for instrumental drifts; the isotope was added to all solutions analyzed at 10 μg/L. Limits of detection (LOD) and limits of quantification (LOQ), calculated as 3 and 10 times the standard deviation of blank measurements, respectively, can be found in a previous publication [82].

### 4.6. Carbon and Nitrogen Determination in Leaves

The percentage of C and N on dry weight was determined by analyzing the dried leaf samples (0.9–1.2 mg) with an elemental analyzer (ECS 4010, Costech, Pioltello, Italy) connected to the ID Micro EA isotope ratio mass spectrometer (Compact Science Systems, Newcastle-Under-Lyme, UK). The same analyses allowed us to determine the C stable isotope composition, δ ^13^C (‰), of the samples. Urea isotopic standard was used to calibrate the instrument. Data were acquired using the software EA IsoDelta (Compact Science Systems, Newcastle-Under-Lyme, UK).

### 4.7. Chlorophyll and Carotenoid Contents 

Sections of 1 cm in length were cut from the center of the third leaf and immediately weighed. Extraction of photosynthetic pigments was carried out using N,N’-dimethylformamide as the solvent, by incubating leaf samples for 7 days at 4 °C in the dark [83]. Concentrations of chlorophyll *a* (Chl *a*), chlorophyll *b* (Chl *b*), and total carotenoids were measured spectrophotometrically with a DU-800 spectrophotometer (Beckman Coulter Inc., Brea, CA, USA), using the extinction coefficients proposed by Porra et al. 1989 [84] and Wellburn 1994 [85], according to the following equations:Chl *a* (µg/mL) = 12.00 A_663.8_ − 3.11 A_646.8_
Chl *b* (µg/mL) = 20.78 A_646.8_ − 4.88 A_663.8_
Carotenoids (µg/mL) = (1000 A_480_ − 1.12 Chl *a* − 34.07 Chl *b*)/245.

Concentrations were calculated and then expressed as mg/g fresh weight (FW).

### 4.8. Chlorophyll a Fluorescence Analysis and JIP-Test

Chlorophyll *a* fluorescence analyses were carried out with a Hansatech Photosynthetic Efficiency Analyzer (Handy PEA, Hansatech Ltd., Norfolk, UK) on the middle region of the third leaf. Light intensity reaching the leaf was 3000 μmol m^−2^ s^−1^, which was sufficient to generate the maximum fluorescence in control and NaCl-treated plants. The fluorescence signals were recorded up to 1 s, as described by Strasser et al. (1995 and 2004) [68,86]. Leaves were dark-adapted for 30 min before determining the minimal fluorescence (F_o_) when all PSII reaction centers are open, and maximal fluorescence (F_m_) when all PSII reaction centers are closed. The O, J, I, and P steps were made visible by plotting the transients on a logarithmic time scale. The OJIP fluorescence transient was doubly normalized at F_o_ and F_m_, and the parameters V_OP_, V_OJ_ and V_IP_ were calculated according to [37,87]. The differences among relative variable fluorescence curves (ΔV_OP_, ΔV_OK_, ΔV_OJ_ and ΔV_IP_) between treated (NaCl) and untreated (Ctrl) plants were calculated. A series of JIP-parameters were derived from the OJIP curves as described by Strasser and co-workers [68,87]. The formulas and definitions of the JIP-test parameters used in the current study are presented in Table 3.

**Table 3 plants-13-00369-t003:** Abbreviations, formulas, and definitions of selected JIP-test parameters derived from the Chlorophyll *a* fluorescence induction curve, based on information presented by Strasser and co-workers [68,87].

Fluorescence Parameters	Description
**Basic parameters calculated from the extracted data**	
F_0_ ≅ F_50μs_ or ≅ F_20μs_	Fluorescence when all PSII RCs are open (≅to the minimal reliable recorded fluorescence)
F_m_ (=F_P_)	Maximal fluorescence, when all PSII RCs are closed (=F_P_ when the actinic light intensity is above 500 μmol(photon) m^−2^ s^1^ and provided that all RCs are active as Q_A_-reducing)
F_V_ = F_m_ − F_0_	Maximal variable fluorescence
Area	Total complementary area between the fluorescence induction curve and the line F = F_P_ (meaningful only when F_P_ = F_m_), relates to the pool size of PSII electron transport acceptors
F_v_/F_0_ = (F_m_ − F_0_)/F_0_	Maximum efficiency of water-splitting reaction on the donor side of PSII
V_t_ = (F_t_ − F_0_)/(F_m_ − F_0_)	Relative variable fluorescence at time t
V_I_ = (F_I_ − F_0_)/(F_m_ − F_0_)	Relative variable fluorescence at the I-step (at 30 ms)
V_J_ = (F_J_ − F_0_)/(F_m_ − F_0_)	Relative variable fluorescence at the J-step (at 2 ms)
M_0_ = [(∆F/∆t)_0_]/(F_m_ − F_0_) = 4 × (F_300μs_ − F_0_)/(F_m_ − F_0_)	Approximated initial slope (in ms^−1^) of the fluorescence transient normalised on the maximal variable fluorescence F_m_ − F_0_ = F_V_; equivalently, initial slope (50 to 300 μs; in ms^−1^) of the V_t_ = f(t) kinetics
S_m_ = Area/(F_m_ − F_0_) = Area/F_V_	Normalised Area (reflecting multiple turnover of Q_A_ reduction events and representing energy necessary for the closure of all RCs)
N = S_m_ × (M_0_/V_J_)	Turnover number (expresses how many times Q_A_ is reduced in the time interval from 0 to tF_m_)
**Specific energy fluxes (per active PSII reaction center)**	
ABS/RC = M_0_ × (1/V_J_) × (1/ϕP_o_)	Absorption flux (exciting PSII antenna Chl *a* molecules) per RC (also used as a unit-less measure of PSII apparent antenna size)
RC/ABS = [(F_2ms_ − F_0_)/4(F_300μs_ − F_0_)] × (F_V_/F_m_) = φP_0_ × (V_J_/M_0_)	Density of RCs per chlorophyll (reciprocal of ABS/RC)
TR_0_/RC = M_0_ × (1/V_J_)	Trapped energy flux (leading to Q_A_ reduction) per RC (at t = 0)
ET_0_/RC = M_0_ × (1/V_J_) × (1 − V_J_)	Electron transport flux (further than Q_A_^−^) per RC (at t = 0)
RE_0_/RC = M_0_ × (1/V_J_) × (1 − V_I_)	Electron flux reducing end electron acceptors at the PSI acceptor side per RC (at t = 0)
DI_0_/RC = (ABS/RC) − (TR_0_/RC)	Dissipated energy flux per RC (at t = 0)
**Quantum yields and efficiencies**	
Ψ_0_ = ET_0_/TR_0_ = (1 − V_J_)	Efficiency/probability that an electron moves further than Q_A_^−^ (at t = 0)
φP_0_ = TR_0_/ABS = [1 − (F_0_/F_m_)] = Fv/F_m_	Maximum quantum yield for primary PSII photochemistry (maximum efficiency at which light absorbed by PSII is used for reduction of Q_A_) (at t = 0)
φE_0_ = ET_0_/ABS = [1 − (F_0_/F_m_)] × Ψ_0_	Probability that an absorbed photon moves an electron further than Q_A_^−^ (at t = 0)
**Performance index**	
PI_(ABS)_ = (RC/ABS) × [φP_0_/(1 − φP_0_)] × [Ψ_0_/(1 − Ψ_0_)]	PSII Performance index on absorption basis

### 4.9. Isolation of Thylakoid Membranes, Gel Electrophoresis and Western Blotting

Fresh harvested leaves were homogenized with a blender in a buffer made of 50 mM HEPES–NaOH (pH 7.8), 0.3 M sucrose, 10 mM NaCl, 5 mM MgCl_2_, 5 mM Ethylenediaminetetraacetic acid (EDTA), and 0.1% (*w*/*v*) bovine serum albumin (BSA). The homogenate was filtered on 8 cotton layers, and centrifuged for 20 min at 3000× *g*. The pellets were resuspended in a buffer made of 50 mM HEPES–NaOH (pH 7.8), 10 mM NaCl, and 5 mM MgCl_2_, and the membranes were spun down at 5000× *g* for 10 min. Finally, thylakoids were resuspended in a buffer made of 50 mM HEPES–NaOH (pH 7.5), 0.1 M sucrose, 10 mM NaCl, and 5 mM MgCl_2_, and the chlorophyll concentration measured according to Lichtenthaler (1987) [88]. All the isolation procedure was performed at 4 °C and avoiding direct light. If not immediately used, thylakoids were frozen in liquid N_2_ and stored at −80 °C.

Thylakoid samples were subjected to SDS-PAGEs performed according to Laemmli’s system [89] on a 12.5% (*w*/*v*) polyacrylamide gel containing 5 M urea. Pre-stained protein size markers (Bio-Rad, Precision Plus, Batavia, IL, USA) were used for estimation of apparent size of thylakoid proteins. The separated proteins were either visualized by staining with Coomassie brilliant blue R-250 or transferred onto nitro-cellulose membrane and immuno-detected with specific antisera against PSI and PSII subunits, by using the alkaline phosphatase conjugate method, with 5-bromo-4-chloro-3-indolyl phosphate/nitro blue tetrazolium as chromogenic substrates (Sigma, Saint Louis, MO, USA). The specific antibodies used in this study were either homemade (PsaA, D1, D2, CP43, CP47, and LHCII; for details, see [90]) or commercially available (PsbO, PsbP, PsbQ, and PsbR; Agrisera codes AS06 142-33, AS06 167, AS06 142-16, and AS05 059, respectively). Proteins were quantified based on the intensities of immunoreactions using the software Quantity One 4.6.1 (Bio-Rad), and relative ratios between intensities of NaCl/Ctrl samples were calculated for each subunit.

### 4.10. Statistical Analyses

Each experiment consisted of at least three independent replicates. Data obtained for all measurements were mean value ± standard deviation (SD). All statistical analyses were performed with the statistical software OriginPro (version 9.0 SR2; Northampton, MA, USA). One-way ANOVA tests were applied to compare salt-treated and control groups, followed by post hoc Tukey’s test (requiring *p* < 0.05).

## 5. Conclusions and Perspectives

From the in-depth morpho-physiological characterization performed on the wild rice *O. rufipogon* and the cultivated *O. sativa* genotypes Baldo (mildly salt-tolerant) and Vialone Nano (salt-sensitive), a higher degree of tolerance to salinity emerged in *O. rufipogon,* as attested by its higher survival rate in salinized conditions. This superior salt tolerance may be related to the ability of *O. rufipogon* to keep the shoot/root biomass ratio constant, accumulate higher amounts of Na^+^ in the roots, thus determining a lower leaf Na^+^/K^+^ ratio, and preserve a higher relative water content in the leaf. All this guarantees this wild rice better preservation of the plant architecture, ion homeostasis, and water status with respect to the cultivated rice genotypes. Moreover, *O. rufipogon,* despite a modest impairment of the photosynthetic performance, preserved the overall photosynthetic apparatus integrity and leaf carbon to nitrogen balance. Conversely, Vialone Nano, after salt treatment, showed the lowest survival rate, and displayed a higher reduction in the growth of shoots rather than roots, with leaves compromised in water and ionic balance, negatively affecting the photosynthetic performance and apparatus integrity. Baldo showed intermediate salt tolerance.

Considering these results, we can suggest that *O. rufipogon* could effectively serve as a source of new genes for pre-breeding towards salt-tolerant lines. In agreement with this statement, there are several reports in which wild progenitors interbreeding with cultivated species represented a source of genetic diversity exploited to develop more resistant and best-performing lines, to cope with biotic or abiotic stressors [91]. The cross-incompatibility between wild and cultivated species is a major hindrance to their effective utilization. However, six of the twenty two wild *Oryza* species belonging to the AA genome, including *O. rufipogon*, are compatible with the cultivated rice, and could play important roles in enhancing the rice’s tolerance to abiotic stresses [92]. For this reason, the demonstrated higher tolerance of *O. rufipogon* to salinity might pave the way for future breeding programs towards the development of salt-tolerant lines. This work represents the first milestone towards the analysis of the genetic determinants of the physiological answer of *O. rufipogon* to salinity and towards the identification of the most salt tolerant accession, considering the availability of thousands of different accessions [93].

Once *O. rufipogon* accessions with valuable levels of salt-tolerance are identified, they can be used in pre-breeding programs to generate introgression lines (ILs), where recurrent parents are represented by *O. sativa* elite varieties/lines with valuable agronomical and quality traits [94]. The ILs can then be used as donors of the salt tolerance traits through breeding, assisted by phenotypic screen (if the genetic loci affecting tolerance have not yet been mapped), or for the mapping of quantitative trait loci (QTLs) involved in salt tolerance. Regarding this last possibility, we have developed a large BC3F4 IL population from the cross between *O. rufipogon* PI 347745 and Vialone Nano, where the last genotype acted as the recurrent parent. This IL population represents a useful genetic resource for the identification of QTLs provided by the wild parent, and polymorphic physiological parameters between tolerant and susceptible genotypes identified in the present work, including the shoot/root biomass ratio and accumulation of Na^+^ in the roots, will address the search for the candidate genes underlying the QTLs. In conclusion, the *O. rufipogon* physiological traits identified here can both be exploited as selection tools during breeding processes, where salt tolerant ILs are used as donor of the trait, and to guide the identification of causal genes identified through a QTL mapping approach.

## Figures and Tables

**Figure 1 plants-13-00369-f001:**
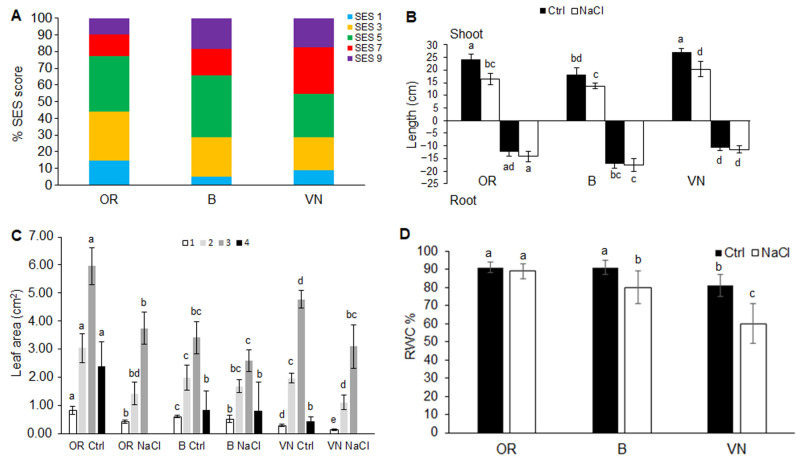
Evaluation of salinity tolerance. SES evaluation after 7 days of exposure to 80 mM NaCl in *O. rufipogon* (OR), *O. sativa* var. Baldo (B), and *O. sativa* var. Vialone Nano (VN) (**A**). In panel (**A**), data expressed as the percentage of plants at each score value (1, highly tolerant; 3, tolerant; 5, moderately tolerant; 7, sensitive; 9, highly sensitive) are the means obtained from three independent experiments (at least 15 plants/genotype each experiment). Shoot and maximal root length (**B**), leaf area (1, first; 2, second; 3, third; 4, fourth leaf) (**C**) and third leaf relative water content (**D**) of plants grown in control conditions (Ctrl) and treated with salt (NaCl). In panels (**B**–**D**), data are the means ± standard deviations of at least eight replicates. For each trait, different letters indicate significant difference determined by one-way ANOVA with Tukey’s test (*p* < 0.05).

**Figure 2 plants-13-00369-f002:**
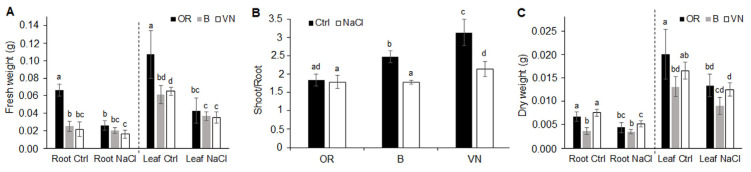
Salt stress effects on the root and leaf biomass. The fresh weight of roots and leaves (**A**), the shoot/root fresh weight ratio (**B**), and the dry weight of roots and leaves (**C**) evaluated in *O. rufipogon* (OR), *O. sativa* var. Baldo (B), and *O. sativa* var. Vialone Nano (VN) grown in control conditions (Ctrl) and after 7 days of salt treatment (NaCl). Data are the means ± standard deviations of at least eight replicates. For each trait, different letters indicate significant differences determined by one-way ANOVA with Tukey’s test (*p* < 0.05).

**Figure 3 plants-13-00369-f003:**
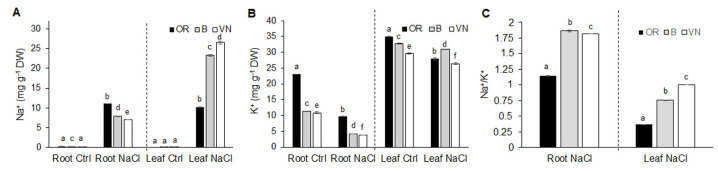
Sodium (**A**) and potassium (**B**) content in roots and leaves of *O. rufipogon* (OR), *O. sativa* var. Baldo (B), and *O. sativa* var. Vialone Nano (VN), grown in control conditions (Ctrl) and after 7 days of salt treatment (NaCl). Na^+^/K^+^ ratio in roots and leaves of salt treated samples (**C**). Data are the means ± standard deviations of three replicates (on pools of at least 30 plants from 3 independent experiments). For each trait, different letters indicate significant differences determined by one-way ANOVA with Tukey’s test (*p* < 0.05). In panel C, Na^+^/K^+^ ratios for control plants are approaching the zero value, and are not shown.

**Figure 4 plants-13-00369-f004:**
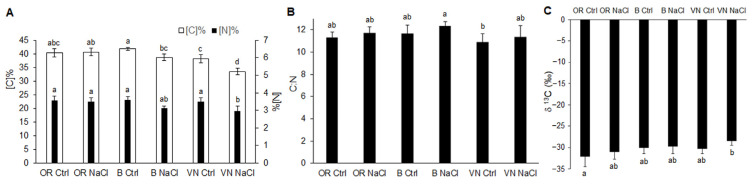
Carbon and nitrogen content (**A**), C:N ratio (**B**), and δ ^13^C (‰) (**C**) in the leaves of *O. rufipogon* (OR), *O. sativa* var. Baldo (B), and *O. sativa* var. Vialone Nano (VN) grown in control conditions (Ctrl) and after 7 days of salt treatment (NaCl). Data are the means ± standard deviations of at least four replicates (on pools of at least 30 plants from 3 independent experiments). For each trait, different letters indicate significant differences determined by one-way ANOVA with Tukey’s test (*p* < 0.05).

**Figure 5 plants-13-00369-f005:**
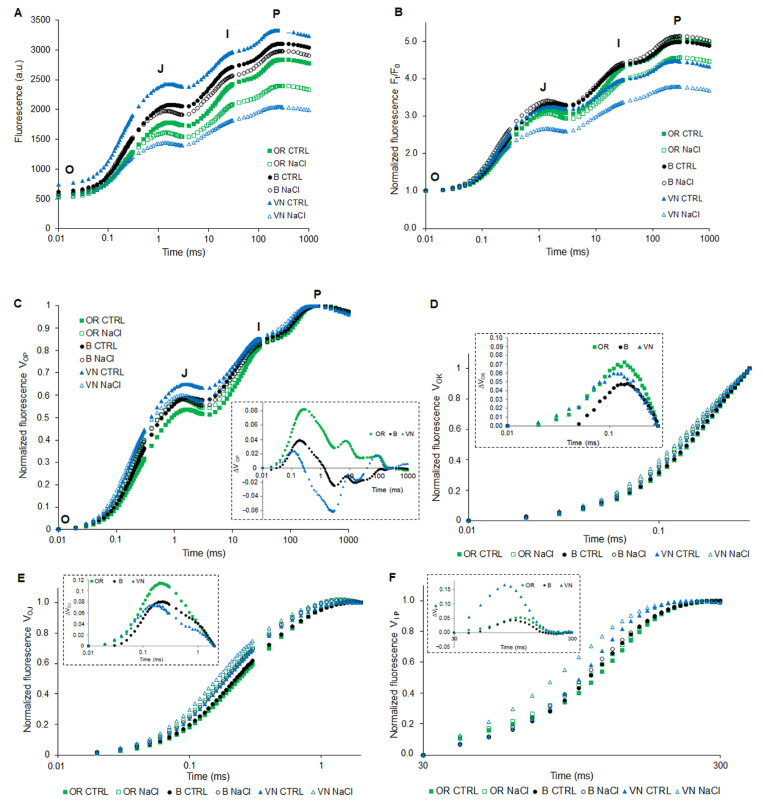
Transient Chlorophyll *a* fluorescence (OJIP-curves) plotted in a logarithmic time scale for fully expanded third leaves obtained from plants of *O. rufipogon* (OR), *O. sativa* var. Baldo (B), and *O. sativa* var. Vialone Nano (VN), grown under control (Ctrl, closed symbols) or after 7 days of salt treatment (NaCl, open symbols). The OJIP curve showed the minimum fluorescence level F_o_ (O) and its maximum F_m_ (P), with intermediate J- and I-steps. Original Chlorophyll *a* fluorescence curves (a.u.; arbitrary unit) (**A**). Chlorophyll fluorescence induction transients normalized to F_o_ (**B**), and normalized to the total amplitude from the O to the P state (V_OP_) (**C**); variable fluorescence V_OK_ (**D**), V_OJ_ (**E**), and V_IP_ (**F**); in the insets, each ΔV = V_NaCl_ − V_Ctrl_. Curves are means of at least 6 replicates for each genotype.

**Figure 6 plants-13-00369-f006:**
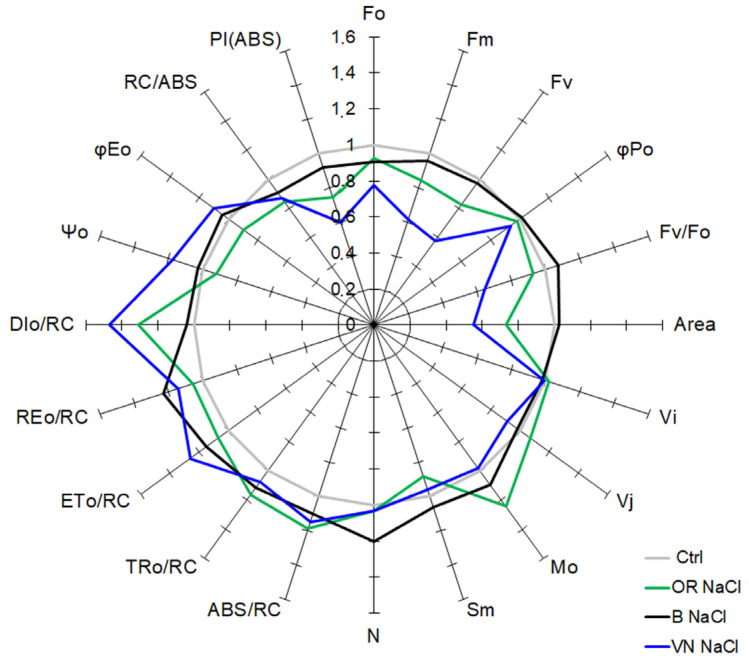
A ‘spider plot’ of selected JIP-test parameters (for definition of parameters, see Table 3) derived from the Chlorophyll *a* fluorescence curves of the untreated (Ctrl) or salt-treated plants (NaCl) shown in Figure 5. All data of JIP-test parameters were normalized to the reference (Ctrl), and each variable at the reference was standardized by giving a numerical value of the unit (one). Data are derived from at least 6 replicates for each genotype.

**Figure 7 plants-13-00369-f007:**
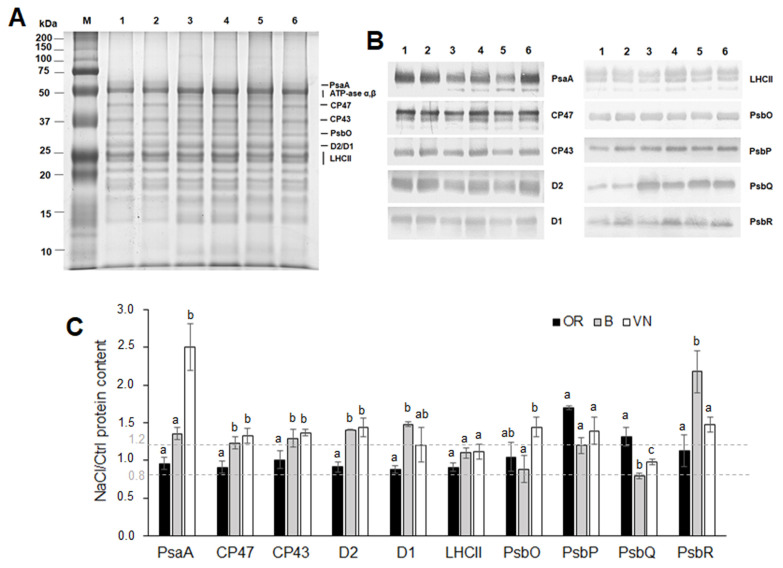
Characterization of thylakoid membranes of plants of *O. rufipogon* (OR), *O. sativa* var. Baldo (B), and *O. sativa* var. Vialone Nano (VN), grown under control (Ctrl) or salt stress for 7 days (NaCl). Coomassie stained SDS-PAGE of thylakoid membranes (**A**), corresponding Western blotting membranes (**B**), and relative protein quantification (**C**) of the main PSI and PSII subunits therein. Lane M, Precision plus protein marker (BioRad); lane 1, OR Ctrl; lane 2, OR NaCl; lane 3, B Ctrl; lane 4, B NaCl; lane 5, VN Ctrl; lane 6, VN NaCl; 4 μg and 1 μg of chlorophyll were loaded on each lane in A and B, respectively. In panel A, protein assessment was performed according to [38]. In panel C, for each subunit in each genotype, data were normalized to the content in the Ctrl sample and the histogram bars represent the average NaCl/Ctrl value calculated from at least three independent blots; for each subunit, different letters indicate significant difference determined by one-way ANOVA with Tukey’s test (*p* < 0.05) among the genotypes.

**Figure 8 plants-13-00369-f008:**
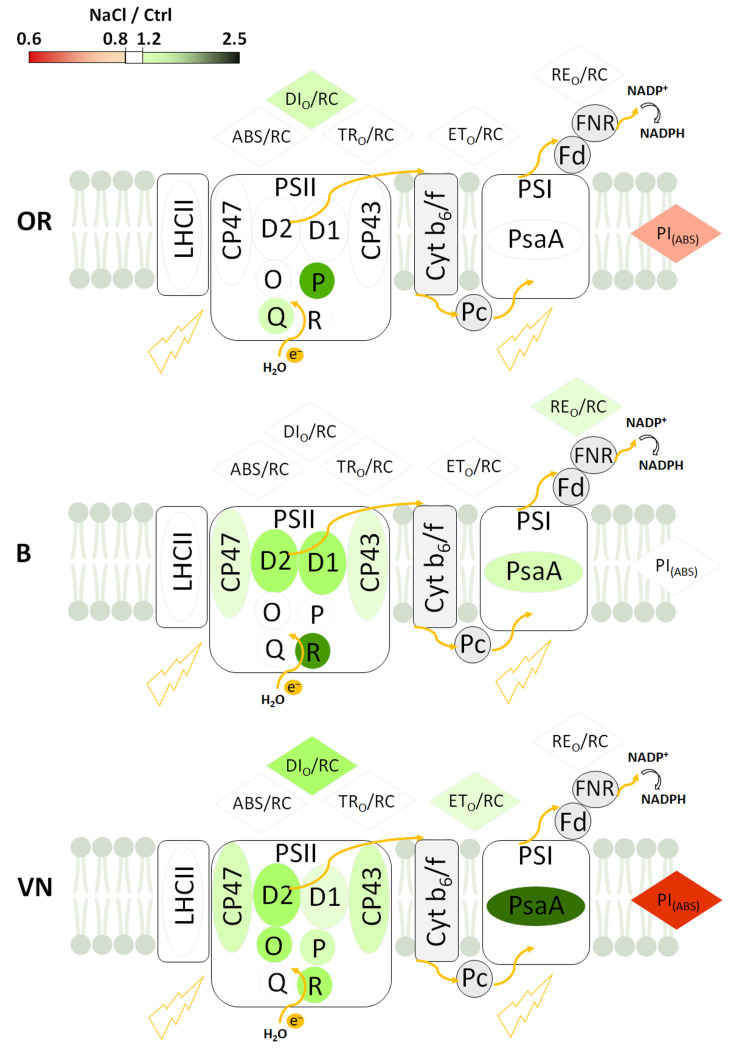
Simplified scheme of the photosynthetic linear electron transport chain summarizing the major detected changes induced by salt stress in the main fluorescence parameters related to energy fluxes and performance index PI_(ABS)_, and in the abundances of PSI and PSII subunits in plants of *O. rufipogon* (OR), *O. sativa* var. Baldo (B), and *O. sativa* var. Vialone Nano (VN). The changes, expressed as NaCl/Ctrl values, refer to data shown in Figure 6 and Figure 7. For each fluorescence parameter/protein subunit abundance, NaCl/Ctrl values between 0.8 and 1.2 are shown in white, below 0.8 in red, and above 1.2 in green. Ovals and circles represent proteins, rhombuses fluorescence parameters. Fd, Ferredoxin; FNR, ferredoxin-NADP^+^ oxidoreductase; O, PsbO; P, PsbP; Q, PsbQ; R, PsbR; Pc, plastocyanin; Cyt b_6_/f, cytochrome b_6_/f.

**Table 1 plants-13-00369-t001:** The effect of salt stress on different root system traits (branching degree, calculated as (Tips − 2)/Length) evaluated in *O. rufipogon* (OR), *O. sativa* var. Baldo (B), and *O. sativa* var. Vialone Nano (VN), grown in control conditions (Ctrl) and after 7 days of salt treatment (NaCl). Data are the means ± standard deviations of at least eight replicates. For each trait, different letters indicate significant differences determined by one-way ANOVA with Tukey’s test (*p* < 0.05).

Plant	Length (cm)	Volume (cm^3^)	Average Diameter (mm)	Surface Area (cm^2^)	Tips Number	Branching Degree
OR Ctrl	295.2 ± 39.6 *a*	0.26 ± 0.04 *a*	0.348 ± 0.030 *a*	29.05 ± 6.06 *a*	520 ±170 *a*	2.14 ± 0.42 *a*
OR NaCl	139.7 ± 32.1 *b*	0.18 ± 0.03 *b*	0.395 ± 0.046 *b*	17.43 ± 3.04 *b*	319 ± 94 *b*	2.25 ± 0.22 *a*
B Ctrl	143.8 ± 25.1 *b*	0.16 ± 0.03 *b*	0.372 ± 0.022 *ab*	17.17 ± 4.3 *b*	375 ± 118 *b*	2.38 ± 0.23 *a*
B NaCl	147.9 ± 24.8 *b*	0.14 ± 0.03 *b*	0.357 ± 0.020 *a*	17.87 ± 3.4 *b*	353 ± 47 *b*	2.41 ± 0.29 *a*
VN Ctrl	240.2 ± 27.9 *a*	0.24 ± 0.03 *ac*	0.357 ± 0.022 *a*	26.04 ± 2.63 *ac*	551 ± 62 *a*	2.30 ± 0.28 *a*
VN NaCl	161.0 ± 27.8 *b*	0.17 ± 0.03 *bc*	0.373 ± 0.025 *ab*	21.04 ± 3.25 *bc*	363 ± 78 *b*	2.37 ± 0.13 *a*

**Table 2 plants-13-00369-t002:** Chlorophyll *a*, chlorophyll *b*, and carotenoid content of *O. rufipogon* (OR), *O. sativa* var. Baldo (B), and *O. sativa* var. Vialone Nano (VN), grown in control conditions (Ctrl) and after 7 days of salt treatment (NaCl). Values are the means ± standard deviations of at least eight replicates. For each trait, different letters indicate significant differences determined by one-way ANOVA with Tukey’s test (*p* < 0.05).

Plant	Chl *a* (mg g^−1^ FW)	Chl *b* (mg g^−1^ FW)	Carotenoids (mg g^−1^ FW)	Chl *a/b*	Chl/Carotenoids
OR Ctrl	3.16 ± 0.35 *a*	0.75 ± 0.05 *a*	0.64 ± 0.07 *a*	4.26 ± 0.32 *a*	6.11 ± 0.18 *ac*
OR NaCl	2.10 ± 0.26 *bc*	0.50 ± 0.07 *bc*	0.41 ± 0.05 *bc*	4.25 ± 0.24 *a*	6.39 ± 0.28 *a*
B Ctrl	2.01 ± 0.27 *bc*	0.45 ± 0.06 *bc*	0.46 ± 0.05 *bc*	4.45 ± 0.26 *ab*	5.42 ± 0.26 *b*
B NaCl	2.31 ± 0.20 *b*	0.52 ± 0.05 *b*	0.48 ± 0.04 *c*	4.47 ± 0.18 *ab*	5.84 ± 0.21 *c*
VN Ctrl	1.96 ± 0.18 *bc*	0.42 ± 0.06 *c*	0.46 ± 0.04 *bc*	4.72 ± 0.14 *b*	5.18 ± 0.24 *b*
VN NaCl	1.89 ± 0.15 *c*	0.43 ± 0.04 *c*	0.40 ± 0.03 *b*	4.45 ± 0.18 *ab*	5.77 ± 0.20 *c*

## Data Availability

Data are contained within the article and Appendix A.

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
