# Peer review of "Physiological Responses to Salt Stress at the Seedling Stage in Wild (Oryza rufipogon Griff.) and Cultivated (Oryza sativa L.) Rice"

_plants, 2024, doi:10.3390/plants13030369_

Round 1

Reviewer 1 Report

Comments and Suggestions for Authors

The manuscript describes that Physiological responses to salinity stress in seedlings of wild rice and cultvation rice. Systemly analzed the phenotypic and physiological changes Under various salt treatments, including root, shoot and leaves traits and their ionic concentration, photosynthetic performance, etc.  I think it could provide some useful information for readers. so it could be accepted after revisions.

1. Figure 1. should be added the phenotypic compared photo under CK-stress?

2. why no molecular level comparison?

3. the format of the references is not consistency, including big letter and small letter.

4. Some key reference should be cited, such as Ren et al. NG, 2005.

5. Key words: wild rice;  and Oryza rufipogon only need one.

Reviewer 2 Report

Comments and Suggestions for Authors

This manuscript titled “Physiological responses to salt stress at seedling stage in wild (Oryza rufipogon Griff.) and cultivated (Oryza sativa L.) rice” presents essential new data about wild and cultivated rice under salt stress. There are several shortcomings to make the manuscript more fruitful.

Some major remarks:

#1: Transcriptome investigation of salt stress responsiveness in seedlings across different cultivars is required for in-depth information.

#2: PSI and PSII regulate salt stress in cultivars; a representative pathway coupled with visual depiction is required.

Some minor remarks:

#1: Along with sample images, PSII "Non-Photochemical Quenching", "Quantum yield of photosynthesis", and "Proportion of active PSII" analysis with graph is required.

#2: Now a days 200mM salt stress introduced in rice seedlings. What are the reasons behind 80mM salt stress?

#3: It will be preaferable to organize the introduction and conclusion including this question:

·         What were the primary objectives of the study?

·         How did O. rufipogon manage ion homeostasis compared to Baldo and Vialone Nano?

·         What role did Na+ accumulation in roots play in the observed differences in salt tolerance?

·         How did salt stress impact the photosynthetic performance of O. rufipogon, Baldo, and Vialone Nano?

·         Why did Vialone Nano show the lowest survival rate, and how was its growth pattern affected by salt stress?

·         What were the implications of the compromised water and ionic balance on the growth of shoots and roots?

·         How does the interfertility of O. rufipogon with O. sativa contribute to its potential as a source of new genes for salt tolerance?

·         What are the practical implications of using O. rufipogon in pre-breeding programs for developing salt-tolerant rice lines?

·         What are the broader implications of the study's findings in terms of improving salt tolerance in cultivated rice?

·         How can the identified physiological traits in O. rufipogon be utilized to enhance salt tolerance in commercial rice varieties?

#4: It will be better if you add a paragraph about “Limitations and Future Perspectives”

Round 2

Reviewer 2 Report

Comments and Suggestions for Authors

Thank you for revising the manuscript. The background transparency of the thylakoid membrane is nearly invisible in Figure 8 (Line No. 646). I would like to suggest that the color scheme might be improved for more visibility.

Author Response

RESPONSE TO REFEREES LETTER

Comment: The background transparency of the thylakoid membrane is nearly invisible in Figure 8 (Line No. 646). I would like to suggest that the color scheme might be improved for more visibility.

Answer: We thank the reviewer for the revision of our work and for this suggestion. We modified the colour of the thylakoid membrane of Figure 8 to improve its visibility.
